# Do Otolaryngologists Over- or Underestimate Laryngopharyngeal Reflux Symptoms and Findings in Clinical Practice? A Comparison Study between the True Prevalence and the Otolaryngologist-Estimated Prevalence of Symptoms and Findings

**DOI:** 10.3390/jcm11175192

**Published:** 2022-09-01

**Authors:** Jerome R. Lechien

**Affiliations:** 1Polyclinic of Poitiers, Elsan Hospital, 86000 Poitiers, France; jerome.lechien@umons.ac.be; 2Department of Anatomy and Experimental Oncology, Mons School of Medicine, UMONS Research Institute for Health Sciences and Technology, University of Mons (UMons), B7000 Mons, Belgium; 3Department of Otolaryngology-Head and Neck Surgery, EpiCURA Hospital, B7000 Baudour, Belgium; 4Department of Otolaryngology-Head and Neck Surgery, EpiCURA Hospital, Rue L. Cathy, University of Mons, B7000 Mons, Belgium

**Keywords:** larynx, laryngitis, laryngopharyngeal, reflux, otolaryngology, head and neck surgery, gastroesophageal reflux, symptoms, signs, overestimation, underestimation

## Abstract

Purpose: To investigate the prevalence of symptoms and signs of laryngopharyngeal reflux (LPR) and to compare them with the otolaryngologist-estimated prevalence of the most common LPR-related symptoms and signs. Methods: The prevalence of LPR symptoms and signs was determined through the clinical data of 403 patients with a positive LPR diagnosis on hypopharyngeal–esophageal multichannel intraluminal impedance pH monitoring. The otolaryngologist-estimated prevalence was assessed through an international survey investigating the thoughts of 824 otolaryngologists toward LPR symptom and sign prevalence. The determination of potential over- or underestimation of LPR symptoms and findings was investigated through a data comparison between the ‘true’ prevalence and the ‘estimated prevalence’ of symptoms and findings by otolaryngologists. Results: The prevalence of breathing difficulties, coated tongue, and ventricular band inflammation was adequately evaluated by otolaryngologists. The prevalence of hoarseness, throat pain, odynophagia, dysphagia, throat clearing, globus sensation, excess throat mucus, tongue burning, heartburn, regurgitations, halitosis, cough after eating or lying down, and troublesome cough was overestimated by otolaryngologists (*p* < 0.01), while the prevalence of chest pain was underestimated as an LPR symptom. Most laryngeal signs, e.g., arytenoid/laryngeal erythema, inter-arytenoid granulation, posterior commissure hypertrophy, retrocricoid edema/erythema, and endolaryngeal sticky mucus, were overestimated (*p* < 0.01). The occurrence of anterior pillar erythema and tongue tonsil hypertrophy was underestimated by participants. Conclusion: Most laryngopharyngeal reflux symptoms and laryngeal signs were overestimated by otolaryngologists, while some non-laryngeal findings were underestimated. Future studies are needed to better understand the reasons for this phenomenon and to improve the awareness of otolaryngologists toward the most and least prevalent reflux symptoms and signs.

## 1. Introduction

Laryngopharyngeal reflux (LPR) is an inflammatory condition of the upper aerodigestive tract tissues related to the direct and indirect effects of gastroduodenal content reflux, which induces morphological changes in the upper aerodigestive tract [1]. The diagnosis is commonly based on symptoms, signs, and the demonstration of hypopharyngeal reflux events on hypopharyngeal–esophageal multichannel intraluminal impedance pH monitoring (HEMII-pH) [1,2]. To date, there is no consensus regarding the standardization of diagnostic criteria. Although HEMII-pH is increasingly recognized as the most reliable diagnostic tool, it is not available in all centers, leading some practitioners to consider symptoms, signs, and a positive response to an empirical therapeutic trial for the LPR diagnosis [3]. However, the non-specificity of symptoms and findings may complicate the diagnosis, because they are prevalent in many otolaryngological conditions, such as rhinosinusitis, allergy, or tobacco-induced pharyngitis [1]. Moreover, it was recently suggested that many otolaryngologist–head and neck surgeons (OTO-HNS) do not consider extra-laryngeal symptoms (i.e., throat pain, odynophagia, halitosis) and findings (pharyngeal erythema, coated tongue, tongue tonsil hypertrophy), which may be prevalent [4,5].

The aim of the present study was to investigate the prevalence of LPR symptoms and signs in a patient cohort and to compare the prevalence outcomes with the otolaryngologist-estimated prevalence of the most common LPR-related symptoms and signs.

## 2. Materials and Methods

### 2.1. Institutional Review Board

Two Institutional Review Board agreements approved the present study. The first was dedicated to the cohort study in which patients consented to participate (CHU Saint-Pierre, Brussels, n°BE076201837630). The second was dedicated to the agreement to conduct the survey study (n°191106).

### 2.2. Inclusion and Exclusion Criteria

From September 2017 to June 2022, 403 patients with LPR-related symptoms, findings, and positive diagnostics at the HEMII-pH were prospectively recruited from the European Reflux Clinic. To be included, patients had to have >1 hypopharyngeal reflux event [5]. The following exclusion criteria were considered: active smoker, alcoholic (>3 alcohol glasses daily), history of upper respiratory tract infection within the last month, neurological or psychiatric illness, head and neck malignancy, head and neck radiotherapy, active seasonal allergies, inhaled corticosteroid intake, or asthma. Patients did not receive antireflux therapy.

### 2.3. Prevalence of Symptoms and Signs

The prevalence of symptoms and signs was based on the patient data. Some clinical data of these patients were used in other previous studies [5,6,7]. Patients were treated with a personalized treatment based on diet, proton pump inhibitors, and alginate [8]. According to a recent review, the LPR diagnosis was based on the occurrence of >1 acid, weakly, or nonacid pharyngeal reflux events [9]. The details about HEMII-pH probe placement and composition were reported in previous studies [5,7].

The prevalence of symptoms and findings was investigated according to the Reflux Symptom Score (RSS) [10] and Reflux Sign Assessment (RSA) [11]. RSS is a self-administered validated 22-item reported-outcome questionnaire evaluating the frequency and severity of otolaryngological, digestive, and respiratory symptoms (Figure 1). The following symptoms were considered in the present study: hoarseness, throat pain, odynophagia, dysphagia, throat clearing, accumulation of throat sticky mucus, globus sensation, tongue burning, heartburn, stomach acid coming up/regurgitations, halitosis, cough after lying down/after meals, troublesome cough, chest pain, and breathing difficulties. RSA is a 61-point validated clinical instrument considering oral, laryngeal, and pharyngeal signs (Figure 2). The following signs were considered: anterior pilar erythema, uvula erythema, coated tongue, oropharyngeal posterior wall erythema, oropharyngeal posterior wall granulations, tongue tonsil hypertrophy, ventricular band erythema/edema, arytenoid or laryngeal erythema, posterior commissure granulations, posterior commissure hypertrophy, retrocricoid edema, and endolaryngeal sticky mucus. The rating of RSA was performed by two board-certified otolaryngologists, one being now retired. RSA is an instrument with a reported adequate interrater reliability outcome (Kendall’s W = 0.663) [11].

### 2.4. Surveyed Prevalence of Symptoms and Signs

The evaluation of the prevalence of symptoms and signs according to the OTO-HNS’ thoughts was based on the findings of a previous international survey, which was developed in an iterative fashion by the LPR Study Group of the Young Otolaryngologists of the International Federation of Oto-Rhino-Laryngological Societies (YO-IFOS) [12]. While the distribution methodology makes it impossible to know how many OTO-HNS received the invitation, we estimate that approximately 8000 otolaryngologists were offered the opportunity to participate. Among them, 824 OTO-HNS (10.3%) determined which RSS symptoms and RSA signs were associated with reflux, leading to an assessment of the ‘estimated prevalence’ of symptoms and signs. The survey was developed with SurveyMonkey^®^ (San Mateo, CA, USA), so that each participant could complete the survey only once. The survey was emailed to members of YO-IFOS; the Confederation of European Oto-Rhino-Laryngological Societies; the European Laryngological Society; and Greek, French, New Zealand, Australian, Korean, Canadian, and Brazilian ear, nose, and throat societies. The responses were collected anonymously. Incomplete responses were excluded from analysis.

### 2.5. Statistical Analyses

Statistical analyses were performed with the Statistical Package for the Social Sciences for Windows (SPSS version 22,0; IBM Corp, Armonk, NY, USA). The proportions of LPR-associated versus non-associated symptoms or findings were compared between groups (patient prevalence versus OTO-HNS-attributed prevalence) with the χ2 test. A *p*-value <0.05 was considered as significant.

## 3. Results

### 3.1. Patient and Participant Features

The features of LPR patients are described in Table 1. There were 228 females (57%) and 174 males (43%). The mean body mass index was 25.2 ± 5.0. Note that 165 patients benefited from gastrointestinal endoscopy, which revealed esophagitis (*n* = 97, 59%), hiatal hernia (*n* = 60, 36%), lower esophageal sphincter insufficiency (*n* = 90, 55%), and gastritis (*n* = 77, 47%). The gastrointestinal endoscopy was normal in 62 cases (38%).

The 824 OTO-HNS who participated to the survey came from Europe (*n* = 264; 32.0%); North America (*n* = 103; 12.5%); East Asia and Oceania (*n* = 129; 15.7%); West Asia and Africa (*n* = 112; 13.6%), and South America (*n* = 216; 26.2%). The responders self-identified as practicing in the following specialties: general otolaryngology (*n* = 472); laryngology (*n* = 190); head and neck surgery (*n* = 170); rhinology (*n* = 166); otology and neuro-otology (*n* = 130), and pediatric otolaryngology (*n* = 126). The survey was completed by 74 residents (9.0%). The board-certified OTO-HNS reported clinical practice experience ranging from 3 to 54 years (mean of 14.6 ± 10.5 years) [4]. A total of 341 participants (41.4%) were identified who used patient-reported outcome questionnaires and sign instruments to evaluate in-office symptoms and signs of LPR.

### 3.2. Prevalence of Symptoms

The statistical analyses identified three types of trends: adequate, significant overestimation, or significant underestimation of symptoms by OTO-HNS (Table 2).

The proportion of OTO-HNS who attributed breathing difficulties to LPR matched with the prevalence (presence of symptoms) of the symptom in the cohort of patients. The comparison between the prevalence of symptoms and the proportion of OTO-HNS who attributed symptoms to LPR reported an overestimation of the following symptoms: hoarseness, throat main, odynophagia, dysphagia, throat clearing, globus sensation, excess throat mucus, tongue burning, heartburn, regurgitations, halitosis, cough after eating or lying down, and troublesome cough. By contrast, the prevalence of chest pain was significantly higher than the OTO-HNS-estimated prevalence (Table 2). The most important differences between the true and estimated prevalence of symptoms occurred for the following symptoms: odynophagia, dysphagia, and tongue burning. For these three symptoms, most patients (>50%) did not report the symptom, while most OTO-HNS attributed these symptoms to LPR (Table 2). The opposite trend was found for chest pain, because most patients (>50%) reported this symptom, whereas a minority of OTO-HNS (<50%) attributed chest pain to LPR. Chest pain was underestimated by OTO-HNS.

### 3.3. Prevalence of Signs

The features of patient signs and OTO-HNS estimations are available in Table 3. The prevalence of coated tongue and ventricular band inflammation matched with the estimated prevalence of OTO-HNS. The following signs were overestimated by OTO-HNS: posterior pharyngeal wall erythema, posterior pharyngeal wall granulations, arytenoid/laryngeal erythema, inter-arytenoid granulation, posterior commissure hypertrophy, retrocricoid edema/erythema, and endolaryngeal sticky mucus. By contrast, the occurrence of anterior pillar erythema and tongue tonsil hypertrophy was underestimated by OTO-HNS (Table 3). The most blatant statistical proportion differences were found for posterior pharyngeal wall erythema, posterior pharyngeal wall granulation, inter-arytenoid granulation, and endolaryngeal sticky mucus. These signs were found in less than 50% of patients, while OTO-HNS attributed them to reflux in more than 50% of cases.

## 4. Discussion

The clinical diagnosis of reflux is a controversial issue due to the non-specificity of symptoms and findings, and the potential related under- or overestimation of LPR symptoms and signs by practitioners [13,14,15,16].

The primary finding of the present study was the demonstration of an overestimation phenomenon of the prevalence of most symptoms and the misevaluation (over- or underestimation) of the prevalence of many signs by OTO-HNS. To the best of my knowledge, this study is the first cross-sectional investigation comparing the prevalence of symptoms and findings with the OTOHNS-estimated prevalence, which limits the comparison with the literature. The overestimation of LPR symptoms and findings by OTO-HNS was previously suggested by Thomas and Zubiaur, who reported an overestimation of LPR diagnosis in patients presenting with hoarseness in the laryngology office [13]. Two main explanations may support this phenomenon.

First, the overestimation of most symptoms of reflux may be related to the high prevalence of these symptoms in out-patients examined in otolaryngology departments. Indeed, the five ‘most prevalent’ symptoms according to OTO-HNS estimation (i.e., hoarseness, throat sticky mucus, throat clearing, globus sensation, and cough) are commonly found in other prevalent otolaryngological conditions, such as chronic rhinosinusitis, allergy, tobacco-induced pharyngitis, and asthma, and related inhaled corticoid treatments [1,17,18]. Thus, the frequent observation of these symptoms may lead to a potential overestimation of their prevalence in LPR, which is known to be prevalent in otolaryngology [1]. Interestingly, the high prevalence of these symptoms led some authors to use the reflux symptom index (RSI) [19] as a clinical tool for the assessment of symptoms and for therapeutic outcomes in some of these conditions [20,21,22].

Second, another hypothesis underlying the overestimation of these symptoms is the consideration of some symptoms but not all LPR-related symptoms in patient-reported outcome questionnaires such as RSI. In the present study, 41% of participants were identified as using patient-reported outcome questionnaires (RSI) or sign instruments (RFS) in daily practice. RSI is the most used worldwide reflux patient-reported outcome questionnaire, and it does not include some prevalent LPR symptoms (e.g., halitosis, throat pain, odynophagia) [10]. The use of RSI in daily practice may lead to the overestimation of the RSI symptoms and the underestimation of symptoms that are not included in the questionnaire. The five ‘most prevalent symptoms’ of LPR regarding OTO-HNS were those of RSI, which may support this second hypothesis. Moreover, most LPR clinical studies conducted over the past few decades used RSI to describe the clinical picture of LPR, which strengthened the association between these symptoms and the disease.

In this study, we observed different trends in the OTO-HNS estimation of signs associated with LPR. Interestingly, the overestimation of signs mainly concerned laryngeal signs (e.g., endolaryngeal sticky mucus, posterior commissure hypertrophy, inter-arytenoid granulation, arytenoid/laryngeal erythema), which are included in the reflux finding score (RFS) [23]. As for RSI, RFS does not consider extra-laryngeal signs such as oral and pharyngeal findings (e.g., anterior pillar erythema, tongue tonsil hypertrophy, coated tongue), which were evaluated as less prevalent by OTO-HNS [11]. The influence of incomplete patient-reported outcome questionnaires and sign instruments on the OTOHNS-estimated prevalence of some symptoms and signs is probably an important issue for future studies. Indeed, the awareness of OTO-HNS toward non-laryngeal signs of LPR remains low [12], while they were assessed as prevalent laryngeal signs in LPR disease [5,11].

The primary limitation of the present study was the low number of survey participants (10.3% of the surveyed OTO-HNS). Indeed, a voluntary survey is vulnerable to sampling error and respondent bias. It is unknown whether the profile of surveyed participants was consistent with the profile and the related LPR knowledge of the general OTO-HNS population. Moreover, the proportion of participants varied from one world region to another, with higher participation of OTO-HNS who came from industrialized countries. The second limitation was the recruitment of LPR patients in only one world region. Indeed, it is theoretically conceivable that the clinical profile of LPR patients may change from one world region to another according to the diet and stress (autonomic nerve dysfunction) differences between world regions, both being favorable factors for LPR development [24,25].

## 5. Conclusions

The majority of laryngopharyngeal reflux symptoms and laryngeal signs are overestimated by otolaryngologists, while some non-laryngeal findings are underestimated. Future studies are needed to better understand the reasons for such a phenomenon and to improve the awareness of otolaryngologists toward the most and least prevalent reflux symptoms and signs.

## Figures and Tables

**Figure 1 jcm-11-05192-f001:**
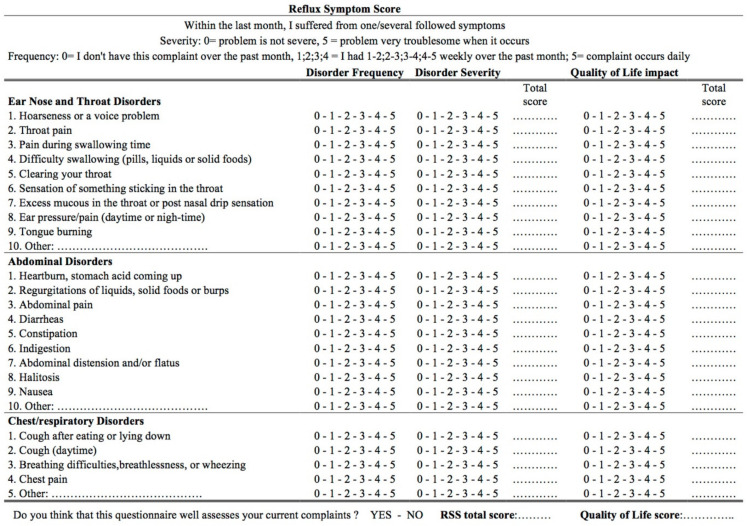
Reflux Symptom Score. The questionnaire is subdivided into three parts according to the complaints: ear, nose, and throat (part 1, 9 items); digestive (part 2, 9 items); and respiratory (part 3, 4 items) symptoms. The frequency and severity of each symptom are rated with a 5-point scale. For each item, the severity score is multiplied by the frequency score to obtain a symptom score ranging from 0 to 25. The sum of these symptom scores is calculated to obtain the RSS final score (ranging from 0 to 550).

**Figure 2 jcm-11-05192-f002:**
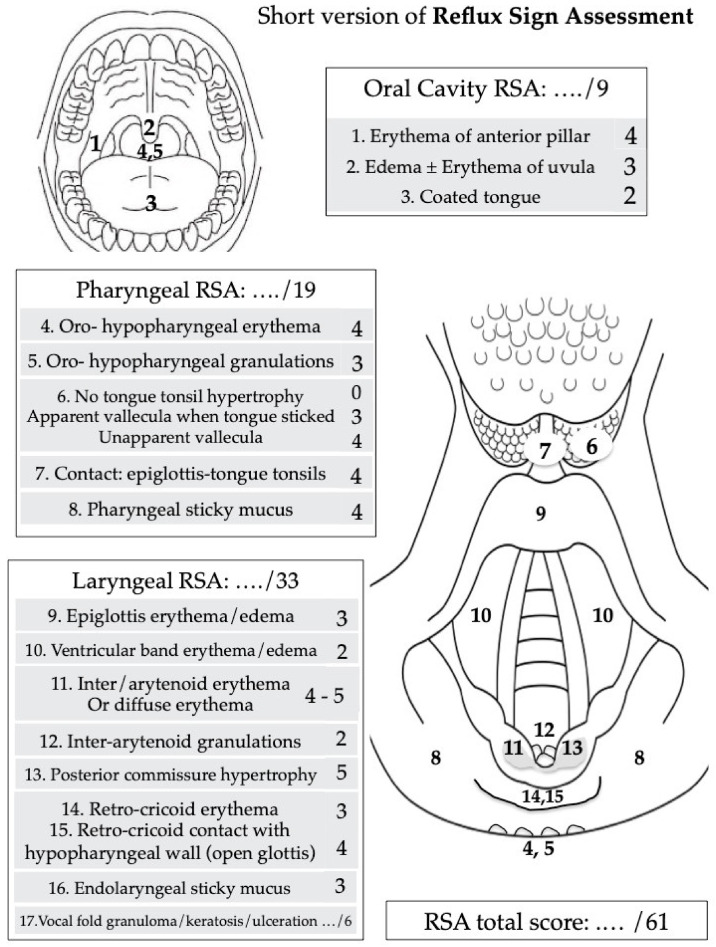
Short version of Reflux Sign Assessment. The tool is subdivided into three parts according to the sign localization: oral cavity, pharynx, and larynx. The occurrence of vocal fold granuloma (+2), keratosis (+2), or ulceration (+2) may be considered in the last item of the score. Due to low prevalence, the following items were removed from the initial version of RSA: edema/erythema of the vocal folds, nasopharyngeal erythema, and subglottic edema/erythema. The total score is calculated by the sum of each item score. The maximum score is 61.

**Table 1 jcm-11-05192-t001:** Epidemiological and clinical features of patients.

Characteristics	*n* = 403
**Age (median, range; years old)**	51.5 (18–90)
**Body mass index (m; SD)**	25.2 ± 5.0
**Gender (*n*, %)**	
Male	174 (43)
Female	228 (57)
**HEMII-pH feature** (m ± SD)	
Pharyngeal acid reflux episodes	12.4 ± 16.5
Pharyngeal nonacid reflux episodes	21.9 ± 42.8
Pharyngeal reflux episodes upright	33.5 ± 46.3
Pharyngeal reflux episodes supine	5.5 ± 11.9
Pharyngeal reflux episodes (total)	34.5 ± 47.9

Abbreviations: HEMII-pH = hypopharyngeal–esophageal multichannel intraluminal impedance pH monitoring; SD = standard deviation.

**Table 2 jcm-11-05192-t002:** Prevalence of symptoms according to patient cohort and otolaryngologists’ thoughts.

Prevalence of Symptoms	Patients	OTO-HNS	*p*-Value
Otolaryngological symptoms			
Voice disorder	61.0	90.3	0.001
Throat pain	65.0	85.0	0.001
Odynophagia	45.9	64.9	0.001
Dysphagia	46.7	57.9	0.001
Throat clearing	77.9	93.8	0.001
Globus sensation	72.2	92.3	0.001
Excess throat mucus	75.9	88.5	0.001
Tongue burning	30.8	59.7	0.001
Digestive symptoms			
Heartburn	72.5	81.2	0.001
Regurgitations or burps	57.6	89.7	0.001
Halitosis	54.8	70.0	0.001
Respiratory symptoms			
Cough after eating/lying down	53.1	97.4	0.001
Troublesome cough	58.6	95.4	0.001
Breathing difficulties	40.9	46.0	NS
Chest pain	56.3	41.4	0.001

Abbreviations: NS = non-significant; OTO-HNS = otolaryngologist–head and neck surgeons.

**Table 3 jcm-11-05192-t003:** Prevalence of signs according to patient cohort and otolaryngologists’ thoughts.

Reflux Sign Assessment Items	Patients	OTO-HNS	*p*-Value
Oral findings			
Anterior pillar erythema	84.9	53.0	0.001
Uvula erythema or edema	36.7	52.1	0.001
Coated tongue	60.5	56.1	NS
Pharyngeal findings			
Posterior oro- or hypopharyngeal wall erythema	38.6	89.1	0.001
Posterior oro- or hypopharyngeal wall inflammatory granulations	28.7	75.8	0.001
Tongue tonsil hypertrophy	81.1	57.4	0.001
Laryngeal findings			
Ventricular band erythema or edema	68.7	70.8	NS
Arytenoid/laryngeal erythema	69.3	97.0	0.001
Inter-arytenoid granulatory tissue	18.1	88.0	0.001
Posterior commissure hypertrophy	76.8	94.5	0.001
Retro-cricoid edema/erythema	59.0	85.5	0.001
Endolaryngeal sticky mucus deposit	38.9	80.4	0.001

Abbreviations: NS = non-significant; OTO-HNS = otolaryngologist–head and neck surgeons.

## Data Availability

Data are available on request.

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
