# Peer review of "Do Otolaryngologists Over- or Underestimate Laryngopharyngeal Reflux Symptoms and Findings in Clinical Practice? A Comparison Study between the True Prevalence and the Otolaryngologist-Estimated Prevalence of Symptoms and Findings"

_jcm, 2022, doi:10.3390/jcm11175192_

Round 1

Reviewer 1 Report

I appreciate the authors' attempt to evaluate the possible over- or underestimation of the laryngopharyngeal reflux (LPR) disease by otolaryngology specialists, since LPR is very common in ENT clinical practice and there is no consensus regarding standardization of diagnostic criteria.

 MAJOR CONCERNS

-        The authors should explain in the Introduction that there is no consensus regarding the standardization of diagnostic criteria, although multichannel intraluminal impedance-pH monitoring (which has been chosen for the study) is increasingly recognized as a diagnostic tool.

-        A paragraph about patient recruitment should be added in the Materials and Methods. The inclusion and exclusion criteria, in fact, should be included in this new sub-paragraph and not in the "prevalence of signs and symptoms" paragraph.

-        Median age of the sample should be added in Table 1.

MINOR CONCERS

-        Please review the manuscript for clarity, redundancy, and typos.

-        References typesetting should be corrected.

Author Response

Editor in chief

J Clin Med

Mons, Mai, 2022

Dear Professor,

I’m sending the paper entitled: “Do Otolaryngologists Over- or Underestimated Laryngopharyngeal Reflux Symptoms and Findings in Clinical Practice? A Comparison Study between the True Prevalence and the Otolaryngologist-estimated Prevalence of Symptoms and Findings." (by Lechien JR.) which is submitted for publication in J Clin Med.

We thank the reviewers for their comments, we considered all of them.

Reviewer 1

I appreciate the authors' attempt to evaluate the possible over- or underestimation of the laryngopharyngeal reflux (LPR) disease by otolaryngology specialists, since LPR is very common in ENT clinical practice and there is no consensus regarding standardization of diagnostic criteria.

Thank you.

 MAJOR CONCERNS

-        The authors should explain in the Introduction that there is no consensus regarding the standardization of diagnostic criteria, although multichannel intraluminal impedance-pH monitoring (which has been chosen for the study) is increasingly recognized as a diagnostic tool.

Thank you. We modified the introduction as requested: Introduction, p.2, line 5: “To date, there is no consensus regarding standardization of diagnostic criteria. Although HEMII-pH is increasingly recognized as the most reliable diagnostic tool, it is not available in all centers, leading some practitioners to consider symptoms, signs and a positive response to an empirical therapeutic trial for the LPR diagnosis [3].”

-        A paragraph about patient recruitment should be added in the Materials and Methods. The inclusion and exclusion criteria, in fact, should be included in this new sub-paragraph and not in the "prevalence of signs and symptoms" paragraph.

We added the requested paragraph: p.2, Methods, paragraph 2:

2.2. Inclusion and exclusion criteria

From September 2017 to June 2022, 403 patients with LPR-related symptoms, findings and positive diagnostic at the HEMII-pH were prospectively recruited from the European Reflux Clinic. To be included, patient had to have >1 hypopharyngeal reflux event [5]. The following exclusion criteria were considered: active smoker, alcoholic (>3 alcohol glasses daily), history of upper respiratory tract infection within the last month, neurological or psychiatric illness, head and neck malignancy, head and neck radiotherapy, active seasonal allergies, inhaled corticosteroid intake or asthma. Patients did not take antireflux therapy.”

-        Median age of the sample should be added in Table 1.

Done. We added: 51.5 yo.

MINOR CONCERS

-        Please review the manuscript for clarity, redundancy, and typos.

Done. We corrected few typos.

-        References typesetting should be corrected.

Done. We corrected.

Reviewer 2 Report

Thank you for the opportunity to review the manuscript titled “Do Otolaryngologists Over- or Underestimated Laryngopharyngeal Reflux Symptoms and Findings in Clinical Practice? A comparison Study between the True Prevalence and the Otolaryngologist Estimated Prevalence of Symptom Findings.”

In this work the author compares ‘true prevalence’ versus ‘estimated prevalence’ of signs and symptoms associated with LPR by comparing clinical and previously published data. 

The strengths of this work include clear writing and thoughtful points in the discussion.  Also, this work brings attention to the need for objective data when diagnosing and attributing specific signs and symptoms to LPR.  

One challenge I had as I read this work was fully understanding how the survey data from citation was framed as 'estimated prevalence' and to fully assess the methods I would have liked to read the original survey questions.  I was unable to locate the original questions.  If they are available perhaps they could be placed in an appendix or cited specifically to assist readers understanding.  

Minor issues:

Title: “underestimated” should be “underestimate” 

Future consideration: Because results reported for the Reflux Sign Assessment (RSA) were completed by the sole author it might be helpful in a future study to demonstrate inter and intra rater reliability of RSA with multiple raters.  This should also be considered a limitation of the current study.  If data have been published regarding the visual perceptual reliability of the RSA please include as a reference.  

Author Response

Editor in chief

J Clin Med

Mons, Mai, 2022

Dear Professor,

I’m sending the paper entitled: “Do Otolaryngologists Over- or Underestimated Laryngopharyngeal Reflux Symptoms and Findings in Clinical Practice? A Comparison Study between the True Prevalence and the Otolaryngologist-estimated Prevalence of Symptoms and Findings." (by Lechien JR.) which is submitted for publication in J Clin Med.

We thank the reviewers for their comments, we considered all of them.

Reviewer 2

Thank you for the opportunity to review the manuscript titled “Do Otolaryngologists Over- or Underestimated Laryngopharyngeal Reflux Symptoms and Findings in Clinical Practice? A comparison Study between the True Prevalence and the Otolaryngologist Estimated Prevalence of Symptom Findings.”

In this work the author compares ‘true prevalence’ versus ‘estimated prevalence’ of signs and symptoms associated with LPR by comparing clinical and previously published data. 

The strengths of this work include clear writing and thoughtful points in the discussion.  Also, this work brings attention to the need for objective data when diagnosing and attributing specific signs and symptoms to LPR.  

Thank you.

One challenge I had as I read this work was fully understanding how the survey data from citation was framed as 'estimated prevalence' and to fully assess the methods I would have liked to read the original survey questions.  I was unable to locate the original questions.  If they are available perhaps they could be placed in an appendix or cited specifically to assist readers understanding.  

Thank you. We added in the appendix the part of the survey dedicated to the evaluation of symptoms and signs by OTOs.

Minor issues:

Title: “underestimated” should be “underestimate” 

We have changed.

Future consideration: Because results reported for the Reflux Sign Assessment (RSA) were completed by the sole author it might be helpful in a future study to demonstrate inter and intra rater reliability of RSA with multiple raters.  This should also be considered a limitation of the current study.  If data have been published regarding the visual perceptual reliability of the RSA please include as a reference.  

The RSA data used in this study were included in other studies and two otolaryngologists rated the RSA (the first author of the present paper and another one who is now retired). Moreover, the RSA appeared to report adequate interrater reliability according to the paper validation (DOI: 10.1177/0003489419888947).

We specified in Methods, p.2, last line: “The rating of RSA was performed by two board-certified otolaryngologists, one being now retired. RSA is an instrument that reported adequate interrater reliability outcome (Kendall’s W= 0.663) [11].”

Thanking you in advance for your attention, I remain,

Best regards,

Jérôme R. LECHIEN, M.D.,Ph.D., M.S.,

Head and Neck surgery, Laboratory of Anatomy and Cell Biology, Faculty of Medicine, University of Mons (UMONS), Avenue du Champ de mars, 6, B7000 Mons, Belgium, [email protected]

Telephone: +32 65 37 35 84

Fax: +32 65 37 31 42

Round 2

Reviewer 1 Report

The authors have done the corrections suggested. The work is now suitable for publication.